# HyperTransformer: Attention-Based CNN Model Generation from Few Samples

## Abstract

In this work we propose *a HyperTransformer*, a transformer based model that generates all weights of a CNN network directly from support samples. This approach uses a high-capacity model for encoding task-dependent variations in the weights of a smaller model. We show on multiple few-shot image classification benchmarks with different model sizes and datasets that our method beats or matches the performance of many traditional learning methods. Specifically, we show that for very small target architectures, our method can generate significantly better performing models than traditional few-shot learning approaches. For larger models we discover that generating the last layer alone allows us to produce competitive or better results while being end-to-end differentiable. Finally, we extend our approach to a semi-supervised regime utilizing unlabeled samples in the support set and further improving few-shot performance in the presence of unlabeled data.

## 1 Introduction

In few-shot learning, a conventional machine learning paradigm of fitting a parametric model to training data is taken to a limit of extreme data scarcity where entire categories are introduced with just one or few examples. A generic approach to solving this problem uses training data to identify parameters $\phi$ of a *learner* $a_\phi$ that given a small batch of examples for a particular task (called a *support* set) can solve this task on unseen data (called a *query* set).

One broad family of few-shot image classification methods frequently referred to as *metric-based learning*, relies on pretraining an embedding $e_\phi(\cdot)$ and then using some distance in the embedding space to label query samples based on their closeness to known labeled support samples. These methods proved effective on numerous benchmarks (see Tian et al. (2020) for review and references), however the capabilities of the learner are limited by the capacity of the architecture itself, as these methods try to build a universal embedding function.

On the other hand, *optimization-based methods* such as seminal MAML algorithm (Finn et al., 2017) can fine-tune the embedding $e_\phi$ by performing additional SGD updates on all parameters $\phi$ of the model producing it. This partially addresses the constraints of metric-based methods by learning a new embedding for each new task. However, in many of these methods, all the knowledge extracted during training on different tasks and describing the learner $a_\phi$ still has to "fit" into the same number of parameters as the model itself. Such limitation becomes more severe as the target models get smaller, while the richness of the task set increases.

In this paper we propose a new few-shot learning approach that allows us to decouple the complexity of the *task space* from the complexity of individual tasks. The main idea is to use the transformer model (Vaswani et al., 2017) that given a few-shot task episode generates an entire inference model by producing all model weights in a single pass. This allows us to encode the intricacies of the available training data inside the transformer model, while still producing specialized tiny models that can solve individual tasks. Reducing the size of the generated model and moving the computational overhead to the transformer-based weight generator, we can lower the cost of the inference on new images. This can dramatically reduce the overall computation cost in cases, where the tasks change infrequently and hence the weight generator is only used sporadically.

We start by observing that the self-attention mechanism is well suited to be an underlying mechanism for a few-shot CNN weight generator. In contrast with earlier CNN- (Zhao et al., 2020) or

BiLSTM-based approaches (Ravi & Larochelle, 2017), the vanilla[1] transformer model is invariant to sample permutations and can handle unbalanced datasets with a varying number of samples per category. Furthermore, we demonstrate that a single-layer self-attention model can replicate a simplified gradient-descent-based learning algorithm. Using a transformer-based model to generate the logits layer on top of a conventionally learned embedding, we achieve competitive results on several common few-shot learning benchmarks. Varying transformer parameters we demonstrate that this high performance can be attributed to additional capacity of the transformer model that decouples its complexity from that of the generated CNN.

We then extend our method to support unlabeled samples by using a special input token concatenated to unlabeled samples to encode unknown labels. In our experiments, we observe that using transformers with two or more layers, we achieve better performance by adding unlabeled data into the support set. We explain our results in Section 4.3 where we show that such a transformer with at least two layers can encode the nearest-neighbor style algorithm that associates unlabeled samples with similarly labeled examples. In essence, by training the weight generator to produce CNN models with best possible performance on a query set, we teach the transformer to utilize unlabeled samples without having to manually introduce additional optimization objectives.

We also show that by generating all layers of the CNN model we can improve both the training and the test accuracies of CNN models below a certain size. The training accuracy can be viewed as a capability of the generated CNN model to adapt to tasks seen at training time, whereas the test accuracy computed on unseen categories characterizes the generalization capability of this model adaptation mechanism. We empirically demonstrate the expected increase of the model training and test accuracies with the increase of the layer size and the number of generated layers (see Figure 3). Interestingly, generation of the logits layer alone appears to be "sufficient" above a certain model size threshold. This threshold is expected to depend on the variability and the complexity of the training tasks. We conjecture that this might reflect the fact that models are sufficiently expressive for the benchmarks we considered, or that additional regularization is needed to prevent overfitting of the meta-learning models.

Finally, in addition to being able to decouple the complexity of the task distribution from the complexity of individual tasks, another important advantage of our method is that it allows to do learning end-to-end without relying on complex nested gradients optimization and other meta-learning approaches, where the number of unrolls steps is large.

The paper is structured as follows. In Section 2, we discuss the few-shot learning problem setup and highlight related work. Section 3 introduces our approach, discusses the motivation for choosing an attention-based model and shows how our approach can be used to meta-learn semi-supervised learning algorithms. In Section 4, we discuss our experimental results. Finally, in Section 5, we provide concluding remarks.

## 2 PROBLEM SETUP AND RELATED WORK

### 2.1 FEW-SHOT LEARNING

The main goal of a *few-shot learning algorithm* is to use a set of training tasks $\mathcal{T}_{\text{train}}$ for finding a *learner* $a_\phi$ parameterized by $\phi$ that given new task domains can train to recognize novel classes using just a few samples per each class. The learner $a_\phi$ can be thought of as a function that maps task description $T = \{(x_i, c_i)\}_{i=1}^t$ containing $k$ labeled input samples $\{x_i, c_i\}$ from $n$ classes, to the weights $\boldsymbol{\theta} = a_\phi(T)$ of a trained model $f(x; \boldsymbol{\theta})$. The parameters $\phi$ are meta-optimized to maximize the performance of the model $f(x; a_\phi(T_S))$ generated using a support set $T_S$ with $x$ drawn from a query set $T_Q$. Each task $T = (T_S, T_Q)$ is randomly drawn from a space of training tasks $\mathcal{T}_{\text{train}}$. Typically, $T_S$ and $T_Q$ are generated by first randomly choosing several distinct classes from the training set and then sampling examples without replacement from these classes to generate $T_S$ and $T_Q$. In a classical "$n$-way-$k$-shot" setting, $n$ is the number of classes randomly sampled in each episode, and $k$ is the number of samples for each class in the support set $T_S$.

The quality of a particular few-shot learning algorithm is typically evaluated using a separate test space of tasks $\mathcal{T}_{\text{test}}$. By forming $\mathcal{T}_{\text{test}}$ from novel classes unseen at training time, we can evaluate

---

[1] without attention masking or positional encodings

generalization of different learners $a_\phi$. Best algorithms are expected to capture the structure present in the training set and to perform well on novel concepts. This structure may, for example, include certain properties of the distributions $p_c(x)$ with $c$ being the class label, or the presence of particular discriminative (or alternatively invariant) features in the tasks from $\mathcal{T}_{\text{train}}$.

## 2.2 RELATED WORK

Few-shot learning received a lot of attention from the deep learning community and while there are hundreds of few-shot learning methods, several common themes emerged in the past years. Here we outline several existing approaches, show how they relate to our method and discuss the prior work related to it.

**Metric-Based Learning.** One family of approaches involves mapping input samples into an embedding space and then using some nearest neighbor algorithm that relies on the computation of distances from a query sample embedding to the embedding computed using support samples with known labels. The metric used to compute the distance can either be the same for all tasks, or can be task-dependent. This family of methods includes, for example, such methods as Siamese networks (Koch et al., 2015), Matching Networks (Vinyals et al., 2016), Prototypical Networks (Snell et al., 2017), Relation Networks (Sung et al., 2018) and TADAM (Oreshkin et al., 2018). It has recently been argued (Tian et al., 2020) that methods based on building a powerful sample representation can frequently outperform numerous other approaches including many optimization-based methods. However, such approaches essentially amount to the "one-model solves all" approach and thus require larger models than needed to solve individual tasks.

**Optimization-Based Learning.** An alternative approach that can adapt the embedding to a new task is to incorporate optimization within the learning process. A variety of such methods are based on the approach called *Model-Agnostic Meta-Learning*, or MAML (Finn et al., 2017). In MAML, $\theta = a_\phi$ is obtained by initializing a DNN at $\theta_0 = \phi$ and then performing one or more gradient descent updates on a classification loss function $L$, i.e., computing $\theta_{k+1} = \theta_k - \gamma \cdot (\partial L/\partial \theta)(T; \theta_k)$. This approach was later refined (Antoniou et al., 2019) and built upon giving rise to Reptile (Nichol et al., 2018), LEO (Rusu et al., 2019) and others. One limitation of various MAML-inspired methods is that the knowledge about the set of training tasks $\mathcal{T}_{\text{train}}$ is distilled into parameters $\phi$ that have the same dimensionality as the model parameters $\theta$. Therefore, for a very lightweight model $f(x; \theta)$ the capacity of the task-adaptation learner $a_\phi$ is still limited by the size of $\theta$. Methods that use parameterized preconditioners that otherwise do not impact the model $f(x; \theta)$ can alleviate this issue, but as with MAML, such methods can be difficult to train (Antoniou et al., 2019).

**Weight Modulation and Generation.** The idea of using a task specification to directly generate or modulate model weights has been previously explored in the generalized supervised learning context (Ratzlaff & Li, 2019) and in specific language models (Mahabadi et al., 2021; Tay et al., 2021; Ye & Ren, 2021). Some few-shot learning methods described above also employ this approach and use task-specific generation or modulation of the weights of the final classification model. For example, in LGM-Net (Li et al., 2019b) the matching network approach is used to generate a few layers on top of a task-agnostic embedding. Another approach abbreviated as LEO (Rusu et al., 2019) utilized a similar weight generation method to generate initial model weights from the training dataset in a few-shot learning setting, much like what is proposed in this article. However, in (Rusu et al., 2019), the generated weights were also refined using several SGD steps similar to how it is done in MAML. Here we explore a similar idea, but largely inspired by the HYPERNETWORK approach (Ha et al., 2017), we instead propose to directly generate an entire task-specific CNN model. Unlike LEO, we do not rely on pre-computed embeddings for images and generate the model in a single step without additional SGD steps, which simplifies and stabilizes training.

**Transformers in Computer Vision and Few-Shot Learning.** Transformer models (Vaswani et al., 2017) originally proposed for natural language understanding applications had since become a useful tool in practically every subfield of deep learning. In computer vision, transformers have recently seen an explosion of applications ranging from state-of-the-art image classification results (Dosovitskiy et al., 2021; Touvron et al., 2021) to object detection (Carion et al., 2020; Zhu et al., 2021), segmentation (Ye et al., 2019), image super-resolution (Yang et al., 2020), image generation

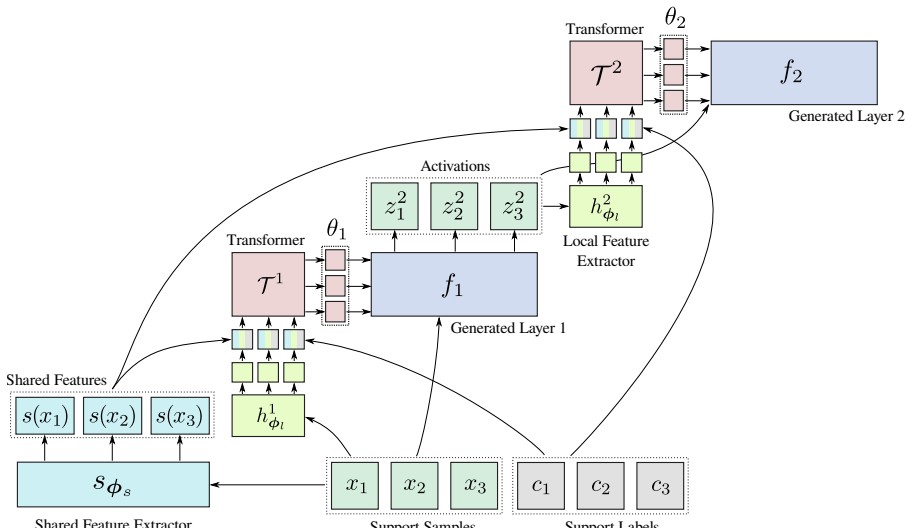

Figure 1: A diagram of our model showing generation of two CNN layers: transformer-based weight generators receive global and local features along with sample labels as their inputs and produce CNN layer weights ($\theta_1$ and $\theta_2$). After being generated, the CNN model is used to compute the loss on the query set. The gradients of this loss are then used to adjust the weights of the entire weight generation model.

(Chen et al., 2021) and many others. There are also several notable applications in few-shot image classification. For example, in Liu et al. (2021), the transformer model was used for generating universal representations in the multi-domain few-shot learning scenario. And closely related to our approach, in Ye et al. (2020), the authors proposed to accomplish embedding adaptation with the help of transformer models. Unlike our method that generates an entire end-to-end image classification model, this approach applied a task-dependent perturbation to an embedding generated by an independent task-agnostic feature extractor. In (Gidaris & Komodakis, 2018), a simplified attention-based model was used for the final layer generation.

## 3 OUR APPROACH

In this section, we describe our approach to few-shot learning that we call a HYPERTRANSFORMER (HT) and justify the choice of the self-attention mechanism as its basis.

### 3.1 FEW-SHOT LEARNING MODEL

A learner $a_\phi$ (as introduced in Section 2.1) is the core of a few-shot learning algorithm and in this paper, we choose $a_\phi$ to be a transformer-based model that takes a task description $T = \{(x_i, c_i)\}_{i=1}^t$ as input and produces weights for some or all layers $\{\theta_\ell | \ell \in [1, L]\}$ of the generated CNN model. For layers with non-generated weights, they are learned in the end-to-end fashion as ordinary task-agnostic variables. In our experiments generated CNN models contain a set of convolutional layers and a final fully-connected logits layer. Here $\theta_\ell$ are the parameters of the $\ell$-th layer and $L$ is the total number of layers including the final logits layer (with index $L$). The weights are generated layer by layer starting from the first layer: $\theta_1(T) \to \theta_2(\theta_1; T) \to \cdots \to \theta_L(\theta_{1,\ldots,L-1}; T)$. Here we use $\theta_{a,\ldots,b}$ as a short notation for $(\theta_a, \theta_{a+1}, \ldots, \theta_b)$.

**Shared and local features.** The *local* features at layer $\ell$ are produced by a convolutional *feature extractor* $h_{\phi_l}^\ell(z_i^\ell)$ applied to the activations of the previous layer $z_i^\ell := f_{\ell-1}(x_i; \theta_{1,\ldots,\ell-1})$ for $\ell > 1$ and $z_i^1 := x_i$. In other words, the transformer for layer $\ell$ receives

$$\mathcal{I}^\ell := \left\{ \left( s_{\phi_s}(x_i), \; h_{\phi_l}^\ell(f_{\ell-1}(x_i, \theta_{1,\ldots,\ell-1})), \; c_i \right) \right\}_{i=1,\ldots,n}.$$

The intuition behind the local feature extractor is that the choice of the layer weights should primarily depend on the inputs received by this layer. The *shared* features, on the other hand, are the same

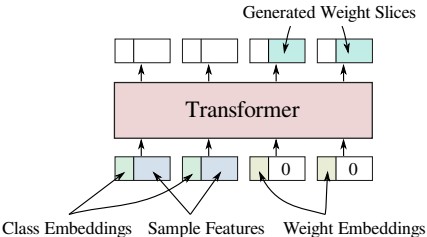

Figure 2: Structure of the tokens passed to and received from a transformer model.

for all layers and are produced by a separate trainable convolutional neural network $s_{\phi_s}(x_i)$. Their purpose is to modulate each layer's weight generator with a global high-level sample embedding that, unlike the local embedding, is independent of the generated weights and is also fully shared between all layer generators.

**Encoding and decoding transformer inputs and outputs.** In our experiments we considered several different architectures of the transformer model, different ways of feeding $\mathcal{I}^\ell$ into it and different ways of converting transformer outputs into the CNN weights.

Input samples were encoded by concatenating local and shared features from $\mathcal{I}^\ell$ to trainable label embeddings $\xi(c)$ with $\xi : [1, n] \to \mathbb{R}^d$. Here $n$ is the number of classes per episode and $d$ is a chosen size of the label encoding. Note that the label embeddings do not contain semantic information, but rather act as placeholders to differentiate between distinct classes.

Along with the input samples, the sequence passed to the transformer was also populated with special learnable placeholder tokens[2], each associated with a particular slice of the to-be-generated weight tensor. After the entire input sequence was processed by the transformer, we read out model outputs associated with the weight slice placeholder tokens and assembled output weight slices into the final weight tensors (see Fig. 2). In our experiments we also considered two different ways of encoding $k \times k \times n_{\text{input}} \times n_{\text{output}}$ convolutional kernels: (a) generating $n_{\text{output}}$ weight slices with each output token having a dimension of $k^2 \times n_{\text{input}}$ (we call it "output allocation"), (b) $k^2$ weight slices of size $n_{\text{input}} \times n_{\text{output}}$ ("spatial allocation"). We show these results in Supplementary Materials.

**Transformer model.** The input discussed above was passed through a sequence of transformer encoder layers and the output weight tokens were then concatenated into a full weight tensor (see Fig. 2). Experiments with alternative architectures employing transformer decoders are discussed in Supplementary Materials.

**Training the model.** The weight generation model uses the support set to produce the weights of some or all CNN model layers. This generated CNN model then processes the query set samples and the cross-entropy classification loss is computed. The weight generation parameters $\phi$ are then learned by optimizing this loss function using stochastic gradient descent.

### 3.2 REASONING BEHIND THE SELF-ATTENTION MECHANISM

The choice of self-attention mechanism for the weight generator is not arbitrary. One motivating reason behind this choice is that the output produced by generator with the basic self-attention is by design invariant to input permutations, i.e., permutations of samples in the training dataset. This also makes it suitable for processing unbalanced batches and batches with a variable number of samples (see Sec. 4.3). Now we show that self-attention can express several intuitive algorithms, thus further motivating its utility.

---

[2]each token is a learnable $d$-dimensional vector padded with zeros to the size of the input sample token

**Supervised learning.** Self-attention in its rudimentary form can implement a method similar to cosine-similarity-based sample weighting encoded in the logits layer[3] with weights $\boldsymbol{W}$:

$$W_{ij} \sim \sum_{m=1}^{n} y_i^{(m)} e_j^{(m)}, \tag{1}$$

which can also be viewed as a result of applying a single gradient descent step on the cross-entropy loss (see Appendix A). Here $n$ is the total number of support-set samples $\{x^{(m)} | m \in [1, n]\}$ and $\boldsymbol{e}^{(m)}$, $\boldsymbol{y}^{(m)}$ are the embedding vector and the one-hot label corresponding to $x^{(m)}$.

The approach can be outlined (see more details in Appendix A) as follows. The self-attention operation receives encoded input samples $\mathcal{I}_k = (\xi(c_k), \boldsymbol{e}_k)$ and weight placeholders $(\mu(i), 0)$ as its input. If each weight slice $W_{i,.}$ represented by a particular token $(\mu(i), 0)$ produces a *query* $Q_i$ that only attends to *keys* $K_k$ corresponding to samples $\mathcal{I}_k$ with labels $c_k$ matching $i$ and the *values* of these samples are set to their embeddings $\boldsymbol{e}_k$, then the self-attention operation will essentially average the embeddings of all samples assigned label $i$ thus matching the first term in $\boldsymbol{W}$ in equation 1.

**Semi-supervised learning.** A similar self-attention mechanism can also be designed to produce logits layer weights when the support set contains some unlabeled samples. The mechanism first propagates classes of labeled samples to similar unlabeled samples. This can be achieved by choosing the queries and the keys of the samples to be proportional to their embeddings. The attention map for sample $i$ would then be defined by a softmax of $\boldsymbol{e}_i \cdot \boldsymbol{e}_j$, or in other words would be proportional to $\exp(\boldsymbol{e}_i \cdot \boldsymbol{e}_j)$. Choosing sample values to be proportional to the class tokens, we can then propagate a class of a labeled sample $\boldsymbol{e}_j$ to a nearby unlabeled sample with embedding $\boldsymbol{e}_i$, for which $\boldsymbol{e}_i \cdot \boldsymbol{e}_j$ is sufficiently large. If the self-attention module is "residual", i.e., the output of the self-attention operation is added to the original input, like it is done in the transformer model, then this additive update would essentially "mark" an unlabeled sample by the propagated class (albeit this term might have a small norm). The second self-attention layer can then be designed similarly to the supervised case. If label embeddings are orthogonal, then even a small component of a class label can be sufficient for a weight slice to attend to it thus adding its embedding to the final weight.

## 4 EXPERIMENTS

In this section, we present HYPERTRANSFORMER (HT) experimental results and discuss the implications of our empirical findings.

### 4.1 DATASETS AND SETUP

**Datasets.** For our experiments, we chose several most widely used few-shot datasets including OMNIGLOT, MINIIMAGENET and TIEREDIMAGENET. MINIIMAGENET contains a relatively small set of labels and is arguably the simplest to overfit to. Because of this and since in many recent publications MINIIMAGENET was replaced with a more realistic TIEREDIMAGENET datasets, we conduct many of our experiments and ablation studies using OMNIGLOT and TIEREDIMAGENET datasets.

**Models.** While HT can support generation of arbitrarily large weight tensors by flattening and slicing the entire weight tensor, in this work, we limit our experiments to HT models that generate slices encoding individual output channels directly. For the target models we focus on 4-layer architectures identical to those used in MAML++ and numerous other papers. Generating larger architectures such as RESNET and WIDERESNET will be the subject of our future work. More specifically, we used a sequence of four $3 \times 3$ convolutional layers with the same number of output channels followed by batch normalization layers, nonlinearities and max-pooling stride-2 layers. All BN variables were learned and not generated[4].

Table 1: Comparison of HT with MAML++ on models of different sizes and different datasets: (a) 20-way OMNIGLOTand (b) 5-way MINIIMAGENET. Results for MAML++ were obtained using GitHub code accompanying Antoniou et al. (2019), those marked with † are from Antoniou et al. (2019). HT outperforms MAML++ on many few-shot tasks. Accuracy confidence intervals: OMNIGLOT – between 0.1% and 0.3%, MINIIMAGENET – between 0.2% and 0.5%.

| Approach | 1-shot (channels) | | | | | 5-shot (channels) | | | | |
|---|---|---|---|---|---|---|---|---|---|---|
| | 8 | 16 | 32 | 48 | 64 | 8 | 16 | 32 | 48 | 64 |
| OMNIGLOT: | | | | | | | | | | |
| - MAML++ | 81.4 | 88.6 | **95.6** | **95.8** | **97.7**† | 83.2 | 94.9 | **98.6** | 98.8 | **99.3**† |
| - HT | **87.2** | **93.7** | 95.5 | 95.7 | 96.2 | **94.7** | **98.0** | **98.6** | 98.8 | 98.8 |
| MINI: | | | | | | | | | | |
| - MAML++ | 43.9 | 46.6 | 49.4 | 52.2† | – | **59.0** | **64.6** | 66.8 | **68.3**† | – |
| - HT | **45.5** | **50.2** | **53.8** | **55.1** | – | 58.5 | 63.8 | **67.1** | 68.1 | – |

## 4.2 SUPERVISED RESULTS WITH LOGITS LAYER GENERATION

Our first experiments compared the proposed HT approach with MAML++ on OMNIGLOT, MINI-IMAGENET and TIEREDIMAGENET datasets (see Table 1). Interestingly, for models that had more than 8 channels per layer, the results obtained with HT generating the final logits layer proved to be nearly identical to those where HT was used to generate all CNN layers (see Section 4.4). In our experiments the number of *local features* was chosen to be the same as the number of model channels and the shared feature had a dimension of 32 regardless of the model size. The shared feature extractor was a simple 4-layer convolutional model with batch normalization and stride-2 $3 \times 3$ convolutional kernels. Local feature extractors were two-layer convolutional models with outputs of both layers averaged over the spatial dimensions and concatenated to produce the final local feature. For all tasks except 5-shot MINIIMAGENET our transformer had 3 layers, used a simple sequence of encoder layers (Figure 2b-i) and used the "output allocation" of weight slices (Section 3.1). Experiments with the encoder-decoder transformer architecture can be found in Appendix D. The 5-shot MINIIMAGENET results presented in Table 1 were obtained with a simplified transformer model that had 1 layer, and did not have the final fully-connected layer and nonlinearity. This proved necessary for reducing model overfitting of this smaller dataset. Other model parameters are described in detail in Appendix B.

Results obtained with our method in a few-shot setting (see Table 1) are frequently better than MAML++ results, especially on smaller models, which can be attributed to parameter disentanglement between the weight generator and the CNN model. While the improvement over MAML++ gets smaller with the growing size of the generated CNN, our results on MINIIMAGENET and TIEREDIMAGENET appear to be comparable to those obtained with numerous other advanced methods (see Table 2). Discussion of additional comparisons to LGM-Net (Li et al., 2019b) and LEO (Rusu et al., 2019) using a different setup (which is why they could not be included in Table 2) and showing an almost identical performance can be found in Appendix C. While the learned HT model could perform a relatively simple calculation on high-dimensional sample features, perhaps not too different from that in equation 1, our brief analysis of the parameters space (see Appendix D) shows that using simpler 1-layer transformers leads to a modest decrease of the test accuracy and a greater drop in the training accuracy for smaller models. We observed that the results in Table 1 could be improved even further by increasing the feature sizes (see Appendix D), but we did not pursue an exhaustive optimization in the parameter space.

It is worth noting that overfitting leading to a good performance on tasks composed of seen categories, but poor generalization to unseen categories, may still have practical applications. Specifically, if the actual task relies on classes seen at the training time, we can generate a model customized to a particular task in a single pass without having to perform any SGD steps to fine-tune the model. This is useful if, for example, the client model needs to be adjusted to a particular set of known classes most widely used by this client. We also anticipate that with more complex data augmentations and additional synthetic tasks, more complex transformer-based models can further improve their performance on the test set and a deeper analysis of such techniques will be the subject of our future work.

---

[3]here we assume that the embeddings $e$ are unbiased, i.e., $\langle e_i \rangle = 0$

[4]Experiments with generated BN variables did not show much difference with this simpler approach.

Table 2: Comparison of MINIIMAGENET and TIEREDIMAGENET 1-shot (1-S) and 5-shot (5-S) 5-way results for HT (underlined) and other widely known methods with a 64-64-64-64 model including (Tian et al., 2020): Matching Networks (Vinyals et al., 2016), IMP (Allen et al., 2019), Prototypical Networks (Snell et al., 2017), TAML (Jamal & Qi, 2019), SAML (Hao et al., 2019), GCR (Li et al., 2019a), KTN (Peng et al., 2019), PARN (Wu et al., 2019), Predicting Parameters from Activations (Qiao et al., 2018), Relation Net (Sung et al., 2018), MELR (Fei et al., 2021). We also include results for CNNs with fewer channels ("-32" for 32-channel models, etc.).

| MINIIMAGENET | | | | | | TIEREDIMAGENET | | |
|---|---|---|---|---|---|---|---|---|
| **Method** | **1-S** | **5-S** | **Method** | **1-S** | **5-S** | **Method** | **1-S** | **5-S** |
| HT | 54.1 | 68.5 | HT-48 | **55.1** | 68.1 | HT-32 | 52.7 | 69.9 |
| MN | 43.6 | 55.3 | SAML | 52.2 | 66.5 | MAML-32 | 51.7 | **70.3** |
| IMP | 49.2 | 64.7 | GCR | 53.2 | **72.3** | HT | **56.1** | **73.3** |
| PN | 49.4 | 68.2 | KTN | 54.6 | 71.2 | PN | 53.3 | 72.7 |
| MELR | **55.4** | **72.3** | PARN | **55.2** | 71.6 | MELR | **56.4** | **73.2** |
| TAML | 51.8 | 66.1 | PPA | 54.5 | 67.9 | RN | 54.5 | 71.3 |

## 4.3 SEMI-SUPERVISED RESULTS WITH LOGITS LAYER GENERATION

In our approach, the weight generation model is trained by optimizing the loss calculated on the query set and therefore any additional information about the task, including unlabeled samples, can be provided as a part of the support set to the weight generator without having to alter the optimization objective. This allows us to tackle a semi-supervised few-shot learning problem without making any substantial changes to the model or the training approach. In our implementation, we simply added unlabeled samples into the support set and marked them with an auxiliary learned "unlabeled" token $\hat{\xi}$ in place of the label encoding $\xi(c)$.

Since OMNIGLOT is typically characterized by very high accuracies in the $97\% - 99\%$ range, we conducted all our experiments with TIEREDIMAGENET. Results of our experiments presented in Table 3 show that adding unlabeled samples leads to a substantial increase of the final test accuracy. Furthermore, notice that the model achieves its best performance when the number of transformer layers is greater than one. This is consistent with the basic mechanism discussed in Section 3.2 and requiring two self-attention layers to function.

It is worth noticing that adding more unlabeled samples into the support set makes our model more difficult to train and it gets stuck producing CNNs with essentially random outputs. Our solution was to introduce unlabeled samples incrementally during training. This was implemented by masking out some unlabeled samples in the beginning of the training and then gradually reducing the masking probability over time.

## 4.4 GENERATING MORE MODEL LAYERS

We demonstrated that HT model can outperform MAML++ on common few-shot learning datasets by generating just the last logits layer of the CNN model. But under what conditions can it be advantageous to generate additional CNN layers? As we show here, generating all CNN layers with a multi-layer transformer can lead to a significant performance improvement on CNN models below a particular size.

Table 3: Test accuracy on TIEREDIMAGENET of supervised 1-shot and 5-shot models and semi-supervised 1-shot models with $u$ additional unlabeled samples per class. The weight generation transformer model uses $L_T$ encoder layers. Notice a performance improvement of semi-supervised learning over the 1-shot supervised results. Accuracy is seen to grow with the number of unlabeled samples and the maximum accuracy is reached when the encoder has at least two layers.

| $(u, L_T)$ | _1-shot_ | _5-shot_ | $(2, 3)$ | $(4, 1)$ | $(4, 2)$ | $(4, 3)$ | $(9, 3)$ |
|---|---|---|---|---|---|---|---|
| Accuracy | 56.0 | 69.9 | 58.3 | 56.6 | 59.9 | 59.9 | 61.5 |

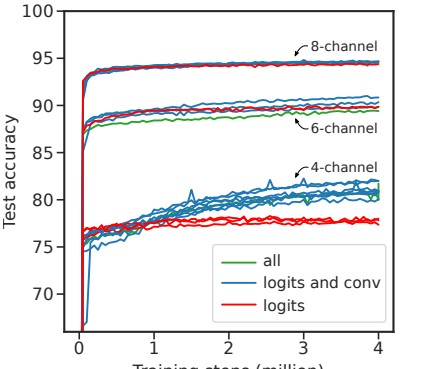 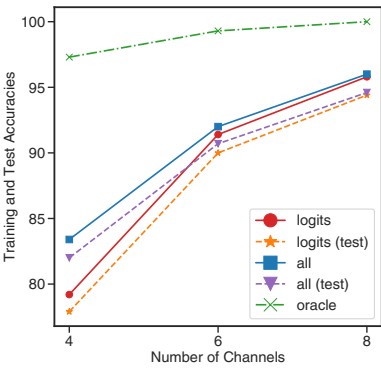

Figure 3: (**Left**) Test accuracies for the generated 4-, 6- and 8-channel CNN models on the 5-shot-20-way OMNIGLOT task. Models with only the last logits layer generated (*red*) are characterized by lower test accuracies compared to the models with some or all convolutional layers also being generated (*blue, green*). Similar plot for TIEREDIMAGENET can be found in Appendix (Fig. 13). (**Right**) 5-shot-20-way OMNIGLOT training/test accuracies as a function of the CNN model complexity: only the final logits layer being generated (*logits*), all layers being generated (*all*), training the model on all available samples for a random set of few classes (*oracle*). A model that generates CNN weights by memorizing all samples (being able to determine their classes) and also memorizing optimal trained weights for any selection of classes would reach the *oracle* accuracy, but would not generalize.

We demonstrated this by conducting experiments, in which some layers were generated and some layers were learned (usually the first few layers of the CNN). For OMNIGLOT dataset, we saw that both training and test accuracies for a 4-channel and a 6-channel CNNs increased with the number of generated layers (see Fig. 3 and Table 4 in Appendix) and using more complex transformer models with 2 or more encoder layers improved both training and test accuracies of fully-generated CNN models of this size (see Appendix D). However, as the size of the model increased and reached 8 channels, generating the last logits layer alone proved to be sufficient for getting the best results on OMNIGLOT and TIEREDIMAGENET.

The positive effect of generating convolutional layers can also be observed in shallow models with large convolutional kernels with large strides where the model performance can be much more sensitive to a proper choice of model weights. For example, in a 16-channel model with two convolutional kernels of size 9 and the stride of 4, the overall test accuracy for a model generating only the final convolutional layer was about 1% lower than the accuracies of the models generating at least one additional convolutional filter. We also speculate that as the complexity of the task increases, generating some or all intermediate network layers should become more important for achieving optimal performance. Verifying this hypothesis and understanding the "boundary" in the model space between two regimes where a static backbone is sufficient, or no longer sufficient will be the subject of our future work.

## 5 CONCLUSIONS

In this work, we proposed *a HyperTransformer* (HT), a novel transformer-based model that generates all weights of a CNN model directly from a few-shot support set. This approach allows us to use a high-capacity model for encoding task-dependent variations in the weights of a smaller model. We demonstrate that generating the last logits layer alone, the transformer-based weight generator beats or matches performance of multiple traditional learning methods on several few-shot benchmarks. More importantly, we showed that HT can be straightforwardly extended to handle unlabeled samples that might be present in the support set and our experiments demonstrate a considerable few-shot performance improvement in the presence of unlabeled data. Finally, we explore the impact of the transformer-encoded model diversity in CNN models of different sizes. We use HT to generate some or all convolutional kernels and biases and show that for sufficiently small models, adjusting all model parameters further improves their few-shot learning performance.

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
