# OpenReview forum: "HyperTransformer: Attention-Based CNN Model Generation from Few Samples"
_ICLR.cc/2022/Conference — ICLR 2022 Submitted_

### Official Review · Reviewer_1KF1 · 2021-11-01

**Correctness:** 3
**Technical Novelty And Significance:** 2
**Empirical Novelty And Significance:** 2
**Recommendation:** 5
**Confidence:** 4

**Main Review:**

1. It is quite interesting to see a self-attention computational circuit can be interpreted as applying a single gradient descent step with the cross-entropy loss.
2. I hate to say, but experimental results are far from state-of-the-art results. why not experimenting with larger CNN and transformers networks?
3. the replacement of existing CNNs with transformers might not be a meaningful contribution to the community.
4. Which encoder architecture did you use eventually? (a) or (b)? I personally prefer to focus on the final architecture you used, and focus on arguing more about it.
5. Any comparisons against other ‘weight generating’ approaches?

[minor comments]
- Fonts in the Figure 1 and 2 are too small, hard to see
- Could you explain about ‘placeholders’ a bit in detail? I think I get the idea, but it's good to be wordy for a broader audience.


**Summary Of The Paper:**

This paper suggests HyperTransformer, a transformer-based model that produces weights of CNN models in meta-learning setup. Experimental results show that the proposed approach improved the performance of CNN models below a certain size in few-shot classification and semi-supervised few shot classification tasks.


**Summary Of The Review:**

This paper shows that the recent transformer architecture works for meta-learning setup as a hypernetwork. Although I believe it is a good information, but I am not fully convinced that this has enough contribution to be a full ICLR paper. Experimental results shows some improvements, but it is not convincing enough so that researchers will follow up on this work later. Therefore, I tend to reject this submission.

---

> ### Author Response · Authors · 2021-11-19
> **Reply to the Reviewer 1KF1**
>
> We would like to thank the reviewer for spending the time to read and review our paper and for providing useful comments that helped us improve the quality of the article.
>
> > “why not experimenting with larger CNN and transformers networks?”
>
> For the model architectures / sizes that we experiment with in the paper, our results are either comparable to, or exceed many state-of-the-art few-shot learning methods (for example, see tieredImageNet, or 1-shot miniImageNet result in Table 1). Some methods that show improvement over our results focus on enriching the training data by generating new training samples, or introducing complex image augmentations, something that was not the goal of our work.
>
> The main reason for why we have not experimented with larger CNN architectures is that the core idea of our approach is to use Transformer-based weight generators with many parameters to generate much smaller CNN models that are specialized to a task. In the paper, we show a significant advantage of this approach over MAML++ on smaller models. The effect expectedly disappears as we increase the size of the generated CNN model. In our preliminary experiments with ResNet-12 architecture, we achieved ~62.7% 1-shot tieredImageNet accuracy (will be included in the paper), which, as we expected, is comparable to other results with the same architecture, but is not improving on them because the size of the model is large enough to accommodate a good “universal” embedding.
>
> > “replacement of existing CNNs with transformers might not be a meaningful contribution”
>
> We are not entirely sure if we understand this comment. The core idea of our work is to use a Transformer model to generate CNN model’s weights. The resulting CNN models are fully standalone and do not need or are replaced by transformers.
>
> > “Which encoder architecture did you use eventually?”
>
> In all the experiments reported in the main text, we used the encoder-only architecture and encoder-decoder results are only reported in Supplementary Materials (SM). The comparison results proved to be inconclusive and a deeper analysis will be the subject of our future work. We agree that mentioning both choices is confusing and will move a discussion of an “encoder-decoder” architecture into SM.
>
> > “Any comparisons against other ‘weight generating’ approaches?”
>
> Even though some methods that we compare with can be interpreted as weight-generation based methods, we will add comparison with other methods that explicitly generate CNN weights. Unfortunately, some weight generation methods use certain choices that make a direct comparison more difficult. For example, LEO (Meta-Learning with Latent Embedding Optimization) conducted ImageNet-based experiments using a pre-trained 28-layer WRN model essentially only generating the last layer on top of sample embeddings. Since we deliberately conducted our experiments generating all layers end-to-end, we could not directly compare with this method. LGM-Net would be another good candidate, but that method uses a non-standard setup that appears to artificially increase their reported accuracy. We will reach out to the authors for clarification, and will try to include comparison with LGM-Net experiments in our updated paper.
>
> > “Could you explain about ‘placeholders’ a bit in detail?”
>
>  We will update the publication with a more detailed explanation of weight placeholders. In models like Vision Transformer, a low-dimensional model output is generated with the help of a special “class token” that is passed to the model along with other inputs. The final output is then read from the position where this token was introduced into the sequence. In a way, this token is used as a sequence element that “accumulates” the output produced by the model. In our application, we need the transformer to generate several outputs that would correspond to different slices of the generated weight tensor. By using specialized learned tokens for each tensor slice, we make sure that the transformer model “knows” where each corresponding weight tensor slice needs to be generated. Another way of saying this is that our “placeholders” essentially act like positional encodings for the slices that the model needs to generate.
>
> > “This paper shows that the recent transformer architecture works for meta-learning setup as a hypernetwork.”
>
> In our opinion, this summary covers only one aspect of our work. On top of showing how transformer models can generate weights for a CNN model, we also explore the advantage of using weight generators in an important regime of small “specialized” generated CNNs.
>
> More importantly, to the best of our knowledge, the proposed method is the first that naturally extends to semi-supervised learning within the same framework (Section 4.3). Our approach allows us to leverage additional unlabeled samples in the support set almost identically to how we use labeled samples, without any additional hand-designed losses or hyper-parameters.

---

> > ### Comment · Reviewer_1KF1 · 2021-11-29
> > **Thanks for the response, here's my final opinion.**
> >
> > I carefully read other reviewers’ opinions and the author’s rebuttal. I really do appreciate all extra experimental results and clarifications that resolved some of my concerns. Also, I understand that the main argument would be to use a ‘larger capacity transformer’ to generate a ‘smaller generated CNN’. Nevertheless, with all due respect, I still think this work is marginally below the acceptance threshold. First, in my humble opinion, the novelty of the proposed approach is limited. The authors used a transformer architecture for a hyper network in the meta-learning setup and I do not think it would be enough contribution to be a full ICLR paper. Second, the proposed approach was only effective for smaller size inference models (the performance improvement is marginal). Although I agree that the smaller size models might be useful in many cases, I think it is a clear limitation. Hence, I will keep my score as it was.

---

> > > ### Author Response · Authors · 2021-11-30
> > > **Thank you**
> > >
> > > We would like to thank the reviewer for the time and effort it took to read the rebuttal and all the reviews. While we agree with an overall description of our method, we would like to point out that the advantage of using HyperTransformer for small models is very significant: in many cases the accuracy increases as high as 5-10 percentage points vs MAML++ (see Table 1). Also, in our opinion, a differentiable few-shot learner with the natural extension to semi-supervised learning (Section 4.3) is innovative and should be interesting to a wide audience.

---

> ### Author Response · Authors · 2021-11-24
> **Summary of new experimental comparisons**
>
> Since the main research question that we tried addressing in our paper was that of "separation into a high-capacity model for parsing the support set, and a low-capacity model for executing" (as Reviewer 2iqR nicely summarized it), we were primarily interested in the case of _small generated CNN models_, where the advantage of representing  the "world knowledge" by the transformer-based weight generator could be fully realized. For the same reason, we did not expect to see a large benefit of using a separate weight generator for large generated CNN models, but it was still informative to compare our performance with that of other recent approaches.
>
> We updated the paper adding comparisons to a weight-generation-based method "Few-Shot Image Recognition by Predicting Parameters from Activations" (Qiao et al., 2018), which has 1-shot and 5-shot miniImageNet accuracies of 54.5 and 67.9, comparable to our accuracies of 54.1 and 68.5 (or 55.1 and 68.1 for a 48-channel HyperTransformer model).
>
> We also replicated experimental parameters used in two other weight-generation approaches called LEO (Li et al., 2019) and LGM-Net (Rusu et al., 2019) and conducted similar experiments adding our results to Appendix C (since they use fairly different setups). In both cases, our results were almost identical to those reported in the original papers. (We will reach out to the authors of LGM-Net to confirm our understanding of their experimental framework.)
>
> Finally, we also included additional Relation Net (Sung et al., 2018) and MELR (Fei et al., 2021) results. We found it interesting that on a larger tieredImageNet dataset, our results were almost identical to those of MELR even though this approach relies on cross-episode relationships.
>
> To address another question, increasing the complexity of the transformer model generally leads to a much better training accuracy. In other words, if our goal is to generate small CNN models that are very well specialized to particular in-distribution tasks (support samples composed of categories seen during training), then increasing the transformer complexity is beneficial. Such generators are also seen to generalize well when generating particularly small CNNs (smaller than 8-channels; see Appendix D, Figure 6). When the generated CNN becomes sufficiently large, much higher transformer complexity can lead to model overfitting and drop in the test accuracy, especially if the dataset is small.
>
> ### miniImageNet
>
> | Model | 1-shot acc. | 5-shot acc. |
> | :-- | :--: | :--: |
> | Predicting Parameters from Activations (64-channel) | 54.5 | 67.9 |
> | MELR (*cross-episode*) | 55.4 | 72.3 |
> | HT (64-channel CNN) | 54.1 | 68.5 |
> | HT (48-channel CNN) | 55.1 | 68.1 |
> | LGM-Net (64-channel) w. rot. augmentations* | 69.1 | 71.2 |
> | HT (64-channel) w. rot. augmentations* | 69.3 | 78.2 |
>
> ### tieredImageNet
>
> | Model | 1-shot acc. | 5-shot acc. |
> | :-- | :--: | :--: |
> | LEO (using LEO embeddings**) | 66.3 | 81.4 |
> | HT (64-channel, using LEO embeddings**) | 66.2 | 81.6 |
> | MELR (*cross-episode*) | 56.4 | 73.2 |
> | Relation Net | 54.5 | 71.3 |
> | HT (64-channel) | 56.1 | 73.3 |
>
> * (*) - The model uses 6 convolutional layers: 4 learned and 2 generated (final). Also, LGM-Net appears to be using random 45-, 90-, etc. degree image rotations (the angle is random and different for different classes, but the same for all images within the same class) at both training and [_test_ stages](https://github.com/likesiwell/LGM-Net/blob/master/experiment_builder.py#L186).
> * (**) - LEO uses a pre-trained WRN-28 model to generate 640-dimensional image embeddings and only generates the final layer; we followed the same approach here (even though we usually generate all layers of the model in the end-to-end fashion without any pre-training).

---

### Official Review · Reviewer_2iqR · 2021-11-01

**Correctness:** 4
**Technical Novelty And Significance:** 3
**Empirical Novelty And Significance:** 3
**Recommendation:** 8
**Confidence:** 4

**Main Review:**

Overall I think the method is fairly interesting and a useful contribution to the "small model" few-shot image classification problem. I think the main weakness of the problem is that peformance seems fairly comparable to existing methods if one is willing to use a "test time" model of any model complexity. While there are several "meta network" approaches that directly generate the weights of another, simpler network, I think there are a few useful/generalizable insights that the paper presents that will be of use to other methods; in particular, the two-stage transformer visualized in Fig 1. The separation into a high-capacity model for parsing the support set, and a low-capacity model for executing, is a natural one for this problem. However in most practical cases I have worked with the cost of a larger backbone network is fairly minimal on modern hardware, it's still nice to have smaller models where possible and the paper demontrates that this can be achieved with comparable accuracy.

When the paper initially claims "the transformer model is invariant to sample permutations" I was slightly confused as this depends on how exactly the transformers are used (in NLP the token order is certainly relevant.)  Some earlier clarity about how the paper intends to use transformers would help here.

Reasserting early on that this is focused entirely on few-shot image classification will also help here, there are plenty of other relevant few shot problems (value regression, image regression/translation, document classification, and so on.)  The paper doesn't clarify this at all in the abstract or early part of the intro.

I am familiar with this general topic but as it is not my primary area of research, I may have missed some prior work approach here.

**Summary Of The Paper:**

The paper proposes a method for solving few-shot image classification that generates all the weights of a very small CNN model. This has the advantage that the generated model can be very small+compact, compared against for example some embedding methods that might require large image classification networks to run as a pre-process. The method is also able to handle unlabeled samples in a fairly natural way.

**Summary Of The Review:**


The paper shows an interesting way to apply "high capacity" transformers to generate the weights of a low-capacity CNN model for few-shot image classification. While results are comparable to SOTA high-capacity classifier networks, the meta-network generation is interesting and able to produce very low-capacity classifiers.

---

> ### Author Response · Authors · 2021-11-15
> **Reply to the Reviewer 2iqR**
>
> We would like to thank the reviewer for spending the time to read and review our paper and for providing useful comments that helped us improve the quality of the article. In the following, we address some of the comments:
>
> 1. _“However in most practical cases I have worked with the cost of a larger backbone network is fairly minimal on modern hardware”_ \
>   We agree that generating a small backbone may not always be useful. However, the size of the model backbone still plays an important role in data-intensive applications like video processing, or in extremely low-power settings such as in embedded controllers. Applying our technique to more complex tasks like image generation or segmentation, where “personalized” models may be much more compact than more generic models, will be the subject of our future work.
>
> 2. _“"the transformer model is invariant to sample permutations" I was slightly confused as this depends on how exactly the transformers are used”_ \
>   We agree with this point and will try to clarify this argument in the text. In the original Transformer model without attention masking (frequently used in autoregressive models), all input tokens are treated identically by the model and any permutation of input tokens results in the identical permutation of model outputs (permutation equivariance). Since in NLP applications, the order of words is important, sentence representations (or similarly, encoding image patches in Vision Transformers) frequently resort to using positional encodings to specify the position of each word in the sentence breaking model equivariance with respect to the word order.
>
> 3. _“Reasserting early on that this is focused entirely on few-shot image classification will also help here”_ \
>   This is a very good point and we will adjust the text accordingly.

---

### Official Review · Reviewer_As45 · 2021-11-02

**Correctness:** 2
**Technical Novelty And Significance:** 2
**Empirical Novelty And Significance:** 2
**Recommendation:** 3
**Confidence:** 4

**Main Review:**

Strengths:

– The overall idea of the paper to increase the learning capacity looks interesting.


Weaknesses:

– I think the novelty of the current version of the paper is incremental, and adding transformer layers between Conv4 would likely increase the Conv4’s capacity due to the over-parameterization.

– While respecting the effort of the author(s) on evaluation of their idea using tieredImageNet, I think we need some additional large datasets instead of Omniglot or miniImagaNet in the case of measuring the effectiveness of a model with higher capacity. Here, deep backbones (such as ResNet) are required in the evaluations. Of course, we should also consider an appropriate regularization technique while increasing the model capacity of deeper backbones.

– The presentation of the paper is not clear in some parts of the paper. For example, there are some grammatical errors in the paper (e.g. opening sentence of section 3). I also had problems with understanding some of the statements. For example, the first sentence of the second paragraph in section 1 “, relies on pretraining to a low-dimensional embedding...“. We know that some metric-based methods (like ProtoNet) can be applied to the flatten output of ResNet12 (resulting in high dimensionality) and still work well.

**Summary Of The Paper:**

This work presents the use of a hybrid CNN-Transformer model for few-shot image classification. Specifically, the paper applies encoding and encoding-decoding transformers between convolution layers to increase the learning capacity.  The paper also includes a Conv-based hybrid model and uses Omniglot, miniImageNet, and tieredImageNet for evaluations.

**Summary Of The Review:**

I think the current version of the paper needs to be strengthened from novelty, evaluation, and presentation perspectives.

---

> ### Author Response · Authors · 2021-11-15
> **Reply to the Reviewer As45**
>
> We thank the reviewer for the time they spent reading and reviewing our paper. We think it’s possible that the main contributions of our paper might have been misunderstood. We note that in our opinion it is a misrepresentation to say that we “add transformers between convolution layers to increase the learning capacity”. Instead, we use a Transformer-based model that receives a small set of support samples as input to generate weight tensors of a CNN inference model that is then used to classify new query-set images. This generated “inference” model _has no Transformer elements_ and retains a simple base CNN architecture, thus we are not actually increasing the complexity of the inference models.
>
> In the following we address specific comments and concerns raised in the review:
>
> * _“I think the novelty of the current version of the paper is incremental”_
>
>   We believe that our approach is novel in many ways. First, we show how to use Transformers to generate an entire CNN model. To the best of our knowledge no such work has been done before. We also demonstrate a significant advantage of using such weight generators vs  MAML-based few-shot learning methods on smaller CNN models.
>
>   Even more importantly, to the best of our knowledge, we are the first to propose a method that has a natural extension to semi-supervised learning within the same general framework (Section 4.3). Our approach allows us to leverage additional unlabeled samples in the support set almost identically to how we use labeled samples, without any additional hand-designed losses or hyper-parameters. This method can be trivially extended even further to samples with partially known labels (when we know that a sample belongs to a certain subset of image categories), or use unlabeled samples from known-to-be-distinct categories (those not present in the support set). We show a basic mechanism relying on a Transformer model with at least two layers that can leverage the presence of unlabeled samples thus further justifying our approach.
>
> * _“adding transformer layers between Conv4 would likely increase the Conv4’s capacity due to the over-parameterization”_
>
>   We are not sure we fully understand this comment and suspect that it could be the result of a misunderstanding of our main contributions. We agree that it is possible to modify the “inference” CNN model architecture by introducing additional model components (thus adding additional parameters). However, the goal of our work was to keep the generated CNN architecture intact and introduce a separate “weight generator” model that would hold all parameters storing the “world knowledge” (knowledge of all training tasks). Such a weight generator will not see samples on which inference is done, instead it will only observe the support set to generate weights. Crucially, such a generator is validated at test time on a new set of labels that were never observed during training.
>
> * _“While respecting the effort of the author(s) on evaluation of their idea using tieredImageNet, I think we need some additional large datasets instead of Omniglot or miniImagaNet in the case of measuring the effectiveness of a model with higher capacity.“_
>
>   We agree that demonstrating the effectiveness of our approach on more complex datasets and larger models is an important direction for further study. Since the transformer-based weight generator can hold a lot more parameters that the generated inference CNN model is using, we expect that increasing data diversity and complexity should improve our model performance, which we will aim to verify experimentally.
>
> * _“Presentation of the paper is not clear.“_
>
>   We agree that the presentation and language of the paper could be improved and will correct all the grammatical and stylistic errors in the paper.
>
> * _“For example, … relies on pretraining to a low-dimensional embedding...”_
>
>   We agree that this was a poor choice of words and will replace “low-dimensional embedding” with just “embedding” in the text.

---

> > ### Comment · Reviewer_As45 · 2021-11-21
> > **Concerned about the evaluations: transformer-based Conv4**
> >
> > While respecting the author's effort on extra clarifications and understanding the use of Conv4 for inference, I still think that the main learner is a transformer-based model which aims at generating the weights of the ConvNet-base feature extractor. Therefore, instead of having "a simple base CNN architecture" as an "inference" model,  I think the inference model is obtained with a hybrid model with additional parameters. Consequently, I have the following concern about the evaluations of the current work with MAML and ProtoNet using only Conv4. The weights of Conv4 are generated with an employed transformer, where the transformer's learning capacity is possibly higher than Conv4.  If this is possible, wouldn't it be better to have a deep backbone such as ResNet12 or ResNet18 to ensure that the classification gains are not simply due to the additional parameters for helping to generate the Conv4's parameters?

---

> > > ### Author Response · Authors · 2021-11-22
> > > **Reply to the Reviewer As45**
> > >
> > > _“wouldn't it be better to have a deep backbone such as ResNet12 or ResNet18 to ensure that the classification gains are not simply due to the additional parameters for helping to build Conv4?”_
> > >
> > > This raised question is one of the key points of the paper. Transformer parameters are most certainly the source of the classification gain. However, in contrast with using deep backbones, these extra parameters are never utilized during inference. The transformer is only fed “training” support set samples (that specify a particular task), while a small generated convolution network is actually used to evaluate new images. If tasks change infrequently, but there are many samples that need to be evaluated for each task (like in video processing), then the cost of running the transformer model can become small compared to the computational savings of using a smaller generated CNN. In other words, our method uses a large number of parameters _only_ for model generation (e.g. per-task learning), while if we were to use deep backbone, we would use all parameters during both per-task learning and _inference_.
> > >
> > > We will update the paper to highlight this point early on.

---

### Decision · Program_Chairs · 2022-01-20

**Decision:**

Reject

**Comment:**

The paper uses a transformer model to generate CNN models and use it for few shot learning.

Although the reviewers appreciate the ideas and the good benchmarking results presented in the paper they are find the paper somewhat incremental compared to previous work in the hyper network literature. This also despite the authors thorough rebuttal with additional results. This shows that the authors could have done a better job in presenting their work.

Rejection is therefore recommended with a strong encouragement to rework the paper to counter future reviewers having similar reservations.